# Muscle Quality Assessment by Ultrasound Imaging of the Intrinsic Foot Muscles in Individuals with and without Plantar Fasciitis: A Case–Control Study

**DOI:** 10.3390/healthcare10030526

**Published:** 2022-03-14

**Authors:** Lorena Canosa-Carro, Daniel López-López, Carmen de Labra, Raquel Díaz-Meco-Conde, Blanca de-la-Cruz-Torres, Carlos Romero-Morales

**Affiliations:** 1Faculty of Sport Sciences, Universidad Europea de Madrid, Villaviciosa de Odón, 28001 Madrid, Spain; lorena.canosa.c@gmail.com (L.C.-C.); raquel.diazmeco@universidadeuropea.es (R.D.-M.-C.); 2Research, Health and Podiatry Group, Department of Health Sciences, Faculty of Nursing and Podiatry, Industrial Campus of Ferrol, Universidade da Coruña, 15403 Ferrol, Spain; daniellopez@udc.es; 3NEUROcom, Centro de Investigaciones Científicas Avanzadas (CICA), Instituto de Investigación Biomé-dica de A Coruña (INIBIC), Faculty of Nursing and Podiatry, University of A Coruna, 15001 La Coruna, Spain; c.labra@udc.es; 4Deparment of Physiotherapy, University of Seville, Avicena Street, 41009 Seville, Spain; bcruz@us.es

**Keywords:** ultrasonography, musculoskeletal diseases, plantar fascia

## Abstract

Objective: The primary aim of the present study was to compare the echo intensity (EI) and echovariation (EV) of the intrinsic foot muscles (IFMs) between individuals with and without plantar fasciitis (PF), using ultrasound imaging. The secondary objective was to study the intra-rater reliability of the echotexture variables. Methods: A case–control study was conducted with 64 participants, who were divided into the following two groups: A, the PF group (*n* = 32); B, the healthy group (*n* = 32). Results: The comparison between the two groups did not identify significant differences (*p* > 0.05) between the flexor hallucis brevis (FHB), flexor digitorum brevis (FDB), quadratus plantae (QP) and abductor hallucis brevis (AHB) variables for the EI and EV. Moreover, excellent intra-rater reliability was reported for the following ultrasound imaging EI variables: ABH (ICC = 0.951), FHB (ICC = 0.949), FDB (ICC = 0.981) and QP (ICC = 0.984). Conclusions: The muscle quality assessment using the EI and EV variables did not identify differences in the FHB, FDB, AHB and QP muscles between individuals with and without PF through USI evaluation. The reliability of all the IFM measurements was reported to be excellent.

## 1. Introduction

At present, plantar fasciitis (PF) is considered to be one of the main foot disorders, with an estimated prevalence of 7% in the general population [1], and among athletes, PF disturbance is more prevalent in runners, being present in up to 17.4% of the running population [2]. According to its etiology, PF has been described as a degenerative soft tissue condition, related to pain, functionality disorders and stiffness alterations in the plantar fascia [3]. Heel pain was the primary symptom reported by the patients in one study [4]. Sub-calcaneal bursitis, plantar nerve disturbances, calcaneal periostitis or heel spur conditions were also frequently reported [5]. Choudhary and Kunal supported the consideration that PF is an overuse injury, due to the repeated trauma etiology [1]. In this context, muscle and soft tissue inflammation episodes could be associated with one another, but their presence remains doubted and understudied [6,7]. Individuals with PF reported that they experience severe pain when they wake up or following non-activity periods (e.g., sleeping or working sitting down) [8]. This condition could develop into a chronic pathology if the symptoms persist over time. Both acute and chronic conditions have been associated with a decrease in quality of life, a lack of functionality and a decrease in sport performance in sport populations [9].

Several authors consider PF to have a multifactorial etiology; for example, a systematic review carried out by Rhim et al. reported a broad range of intrinsic and extrinsic risk factors in the context of PF [10,11,12]. The intrinsic risk factors reported were as follows: An excessive body mass index (BMI), reduced toe plantar flexion and restricted ankle dorsiflexion, reduced eversion and inversion mobility, Achilles and tibialis tendon disorders, excessive pronation, pes planus or cavus and weakness or disturbances in the intrinsic and plantar foot muscles [11]. The extrinsic risk factors reported were as follows: excessive physical activity, poor-quality footwear, inadequate surfaces or even walking barefoot [12].

Intrinsic foot muscles (IFMs) play a key role in providing movement and stability to the ankle and foot; for example, they act as a support for the foot arches [13]. IFMs, such as the abductor hallucis brevis (AHB), flexor digitorum brevis (FDB) and quadratus plantae (QP), also coordinate with the extrinsic foot muscles, in order to transmit force and mobility to the foot, and to modify foot stiffness [14]. Therefore, these soft tissue structures have been considered a target for study and assessment, due to their importance in the diagnosis and management of ankle and foot disorders.

Lately, the ultrasound imaging (USI) technique has been employed for the assessment of ankle and foot structures; for example, Wu et al. argued that ultrasonography is a reliable method for the evaluation of the plantar fascia [15]. Architectural changes, such as the thickness of the plantar fascia, were explored in subjects with chronic non-insertional Achilles tendinopathy who exhibited a shorter plantar fascia upon insertion [16]. In relation to the IFMs, Calvo-Lobo et al. reported a decrease in the thickness of the IFMs in patients who had suffered a stroke [17]. The cross-sectional area (CSA) and thickness were also decreased in the AHB and FHB muscles in individuals who had been diagnosed with hallux valgus [18]. In regard to foot disorders, such as pes pannus, Angin et al. reported an increase in the thickness of the plantar fascia in patients with this foot condition, with overpronation being an associated risk factor for PF or Achilles tendinopathy pathologies [19]. Several authors support the use of USI for the assessment of the IFMs, considering its quick examination procedure, relatively cheap cost and portable features, which make it a suitable tool for clinicians and physical therapy researchers [20]. Recently, there has been exponential growth in the use of USI for the analysis of muscle quality due to its applicability. Soft tissue and muscle disturbances have been related to an increase in stress and changes in the mechanical loads compromising the texture image quality. Thus, ultrasound descriptors have been used for the analysis of muscle quality parameters, based on the gray-scale pixel distribution [21,22]. The main echotexture parameters are as follows: echo intensity (EI), described as the mean of the pixel gray-scale intensity, and echovariation (EV), described as the statistical parameter used to assess the pixel distribution dispersion, which employs the mean and the standard deviation (SD) [23]. In this context, several authors have employed these qualitative ultrasound descriptors for the assessment of muscle and soft tissues [24,25,26]. Muscle tissue is composed of contractile and non-contractile tissue (e.g., adipocytes) and could be involved in the function of the muscle. In line with this, several authors have argued that muscles that present atrophy or soft tissue pathology report higher EI values [27]. Therefore, the primary aim of the present study was to assess the intra-rater reliability of the echotexture variables. The secondary objective was to compare the EI and EV of the IFMs between individuals with and without PF, using ultrasound imaging. We hypothesized that patients diagnosed with PF might compensate for the plantar fascia deficit with an overcompensation of the IFMs with fascial and muscle architecture modifications due to a load adaptation process. Thus, patients with PF might exhibit differences in the pixel gray-scale intensity of the IFMs, assessed by USI. In addition, we hypothesized that the intra-rater assessment of the muscle quality of the IFMs will show that it is a reliable method.

## 2. Methods

### 2.1. Study Design

The study design comprised a cross-sectional investigation carried out from August to September 2021. The present study was a secondary analysis from a prior research about quantitative ultrasonography variables [28]. Information on this study was reported in adherence to the STROBE checklist [29]. The Ethics Committee of the Universidad Europea approved the study (October 2020, code: CIPI/20/166). All participating patients provided written informed consent.

### 2.2. Participants

A total of 64 volunteers, aged 18 to 55 years old, were enrolled for the present study. The sample was divided into the following two groups: the PF group (A) (*n* = 32) and the healthy group (B) (*n* = 32). Inclusion criteria for the PF group were as follows: heel pain for at least 1 month, tenderness and discomfort in the middle of the plantar fascia or in the medial calcaneal tubercle during palpation and immediate pain upon waking up or after non-weight-bearing activities, such as working in a seated position [30,31]. Inclusion criteria for the control group were as follows: healthy participants with no PF diagnosis and no other foot or ankle condition or symptomatology. In both groups, subjects were excluded if they presented with the following: lower limb disorders, lower back pain, any bone condition, systemic disease, infection, plantar orthoses or a discrepancy greater than 1 cm in length, or if they were undergoing corticosteroid treatment [32]. The enrollment of participants was carried out by a medical doctor, and the diagnosis was based on symptomatology and the clinical history of the patient.

The calculation of the sample size was developed by a prior analysis for the FHB CSA variable in a pilot study (*n* = 10), with an effect size of 0.86, a one-tailed hypothesis, a power of 0.80 and an α error of 0.05. Thus, a total sample of 64 participants was calculated. G*Power software was employed to develop the sample size calculation.

### 2.3. Procedure

At the beginning of the study, all participants were physically examined by an experienced physical therapist to confirm their suitability for the study. Height and weight variables were measured for all patients, and the body mass index (BMI) was also calculated.

### 2.4. Outcome Measurements

A high-quality Mindray DC-60 with a 6 to 14 MHz linear transducer (L14-6NE) was used to measure all the variables. IFM thickness variables were recorded following the guidelines reported by Mickle et al., with subjects placed in the supine position with the knee slightly flexed [33]. The thickness of the FHB was assessed by placing the transducer longitudinally to the 1st metatarsal bone at the thickest section of the muscle. For the thickness examination of the FDB and QP muscles, the transducer was placed longitudinally along a line from the medial calcaneal tubercle to the third toe on the thinner section of the muscle. With the transducer in the same place as for the FDB muscle, the thickness of the QP muscle was evaluated immediately under the aforementioned structure (Figure 1). For the AHB assessment, the transducer was placed at the middle of the line between the origin of the muscle on the calcaneal tuberosity and the navicular tuberosity [33] (Figure 2). For each variable, final scores were calculated with ImageJ software (Bethesda, MD, USA), using the mean of 3 repeated measurements. Following previous guidelines, the EI and EV variables were extracted from a region of interest (ROI) of 71 × 40, with an 8-bit gray scale. Muscle quality variables were examined and quantified based on the pixel histogram. EI was described as the mean value of the gray-scale pixel distribution, and EV was defined as the relation between the EI and the SD of the pixel distribution (EV = (SD/mean) × 100) [34].

### 2.5. Statistical Analysis

SPSS v.22 (IBM, Armonk, NY, USA) was employed to carry out the statistical analysis, with an alpha error of 0.05, considering a *p*-value < 0.05 to be statistically significant, with a 95% confidence interval (CI). The Kolmogorov–Smirnov test was used to check the normality assumption. The mean and standard deviation (SD) were calculated, and Student’s *t*-test was performed for parametric data, while the median, interquartile range (IR) and Mann–Whitney *U* test were employed for non-parametric data. Additionally, Levene’s test was used to check the equality of variances. The intra-class correlation coefficient (ICC) was developed in order to analyze the intra-rater reliability between trials.

Subsequently, in order to reduce bias for grouping selection, a linear regression model was run for each independent sociodemographic variable and each dependent ultrasonography variable.

## 3. Results

The descriptive data are reported in Table 1 for the PF (*n* = 32) and healthy groups (*n* = 32), and they showed significant differences (*p* < 0.05) in age, weight and body mass index (BMI) between the two groups. The comparison between the outcome measurement differences between the groups showed that the differences were not significant (*p* > 0.05) for the FHB, FDB, QP and AHB variables for the EI and EV (Table 2) (Figure 3). Excellent intra-rater reliability was reported for the ultrasound imaging of the following EI variables: ABH (ICC = 0.951), FHB (ICC = 0.949), FDB (ICC = 0.981) and QP (ICC = 0.984).

The linear regression results show non-significant differences for all of the dependent variables checked (*p* = 0.102 to *p* = 0.789). Thus, there was no bias in sociodemographic variables in either group.

## 4. Discussion

This study assessed and compared the echotexture variables of the IFMs in individuals with and without PF. Contrary to our hypothesis, the results show that there was no significant difference in the EI and EV variables between the two groups. However, the reliability of the measurements was excellent. To the best of our knowledge, the present study is the first study regarding the assessment of the muscle quality of the IFMs using USI. No significant differences were found for the EI and EV variables in the FHB, FDB, HB and QP muscles between the PF and healthy groups.

Previous research has highlighted the importance of the IFMs in the biomechanics of the foot and ankle; these muscles have origins and insertions that span the length of the longitudinal arch [35]. Therefore, these structures play an important role in the gait phases. Cheung et al. speculated that a weak IFM directly affects the plantar fascia due to insufficient longitudinal arch support which produces an excessive strain at the plantar fascia insertion and mid-portion [36]. The authors remarked that atrophy of the IFMs might be associated with PF disturbances. Based on previous research, the assessment of the IFMs could provide useful information for diagnostic and rehabilitation strategies in patients with PF [10,37,38].

USI tissue characterization demonstrates a more disorganized architecture muscle pattern increasing the EI due to an increase in fibrous muscle infiltration and connective tissue changes [39]. In this line, Smith et al. argued that muscle tissue assessment and characterization are fundamental to investigating the linkages between the muscle features and biomechanics [40]. The ultrasonography echotexture values obtained for the FHB, FDB, AHB and QP muscles did not show significant differences for the EI and EV variables between the PF and healthy groups.

Regarding quantitative ultrasound evaluation, previous work has shown values of the IFMs in lower limb conditions: for example, in individuals diagnosed with chronic non-insertional Achilles tendinopathy, who exhibited an increase in the thickness of the FDB, as well as the FDB and FHB CSA [41]. In relation to the IFMs and the center of pressure, Zhang et al. suggested that a larger AHB, evaluated by USI, correlated with a smaller center of pressure during single-leg standing [42]. Other studies have shown that larger IFMs are related to a high maximum force and contact area of the foot, which are known to be found in foot disorders, such as PF [43]. Morikawa et al. showed that the ultrasonography muscle architecture of the FHB and the plantar fascia were related to the force attenuation in the single-drop jump, and that the AHB and the plantar fascia were related to the repetitive rebound jump test [44]. These results suggest that changes in the muscles and soft tissues, developed by incorrect biomechanics of the ankle and foot, or heavy load fluctuations, may be observed using USI.

To date, qualitative ultrasonography methods for characterizing muscle tissue have revealed fibrosis and fatty infiltration in individuals with musculoskeletal disorders compared with individuals without musculoskeletal disorders, based on the average gray-scale analysis which provides more objective information [45]. However, based on the findings of the present study, it has not yet been proven that these changes are sufficient to modify the pixel gray scale, as evidenced by the results of the echotexture variables. In our study, ROI selection and ultrasonography procedures were developed according to prior research in order to ensure that selected areas presented maximum brightness. Although the quality ultrasound examinations did not report differences in the IFMs, the tendency showed higher values for the control group, which may suggest that in patients diagnosed with PF (minor EI and EV), less homogeneity, with the presence of a hypoechoic ultrasonography pattern, may be reported. Coinciding with the results of the present study, Baellow et al. did not find differences in the muscle quality of the IFMs in individuals with patellofemoral pain with respect to healthy controls [46].

Regarding the qualitative USI assessment, Martinez-Payá et al. observed differences in the EV parameter in the biceps brachialis, tibialis anterior and quadriceps femoris in patients with amyotrophic lateral sclerosis (25). In the same context, Almazán-Polo et al. did not observe significant differences in the EV parameter in individuals with lumbopelvic pain, with respect to healthy subjects [47]. At present, muscle quality parameters are being proposed as progression or prognostic biomarkers that may aid clinicians and researchers in showing interesting and useful results through clinical applications; however, more studies are required to provide deeper knowledge on this subject, and to establish USI examination protocols.

## 5. Clinical Applications

The assessment of muscle and soft tissue quality in B-mode USI, whilst studying the pixel intensity, could be useful. Thus, evaluating the IFM echotexture in subjects with PF may provide additional information about the fatty muscle infiltration and soft tissue features which could be related to the contraction capacity and contractile features of the muscle. The present study reported that patients diagnosed with PF did not present higher EI values than the controls, suggesting that the IFMs did not present atrophy or neural disturbances. In addition, the quality assessment, combined with the quantity evaluation by USI and the clinical orthopedic test, should be considered a priority for the diagnosis and management of patients with PF. However, more studies are still necessary in this field of research to expand the knowledge about ultrasonography quality assessment and the neuromusculoskeletal system in individuals with and without PF.

## 6. Methodological Considerations

Some limitations of the present study should be acknowledged. Firstly, the main limitation was that the USI assessment was not performed by a blinded operator, which could increase the risk of bias. Secondly, there were differences in the BMI between the two groups, and this must be considered in the interpretation of the results, given that the BMI has been associated with hypertrophy of the plantar fascia and intrinsic foot muscles (38). Finally, neural disturbances directly affect the musculoskeletal system, and no neural tests were developed in the present study. Future studies should match age and BMI between both groups. In addition, future research with novel ultrasound systems, such as elastography or M-mode, are needed to assess soft tissue stiffness and mobility.

## 7. Conclusions

The muscle quality assessment using the EI and EV variables did not report significant differences in the FHB, FDB, AHB and QP muscles between individuals with and without PF, evaluated using USI. Nevertheless, excellent reliability was reported for all the IFM measurements. Evaluation of the muscle quality of the musculoskeletal tissue could aid clinicians and researchers, but more studies are required to provide a deeper understanding, and a solid base to establish USI examination protocols.

## Figures and Tables

**Figure 1 healthcare-10-00526-f001:**
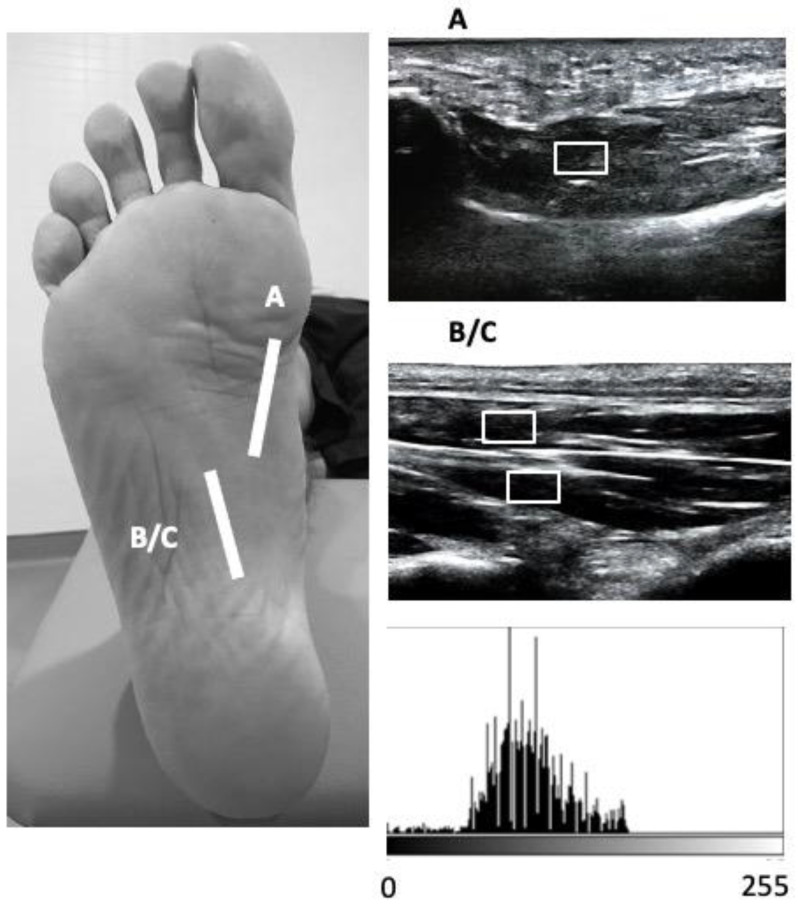
Ultrasound imaging of the FHB, FDB and QP in the longitudinal view.

**Figure 2 healthcare-10-00526-f002:**
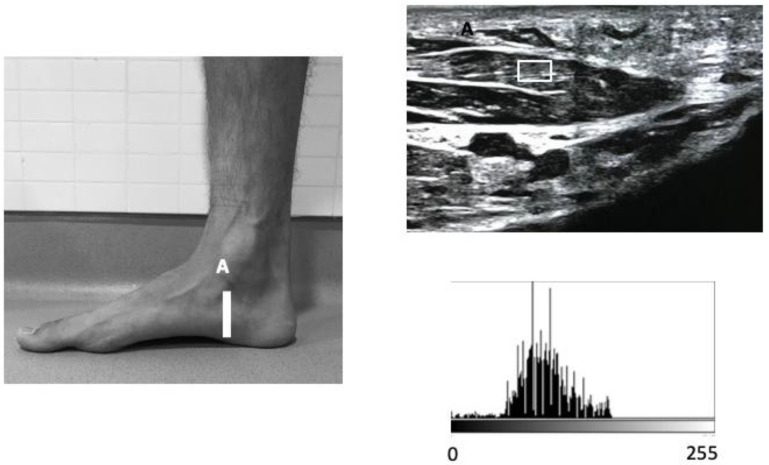
Ultrasound imaging of the AHB in the longitudinal view.

**Figure 3 healthcare-10-00526-f003:**
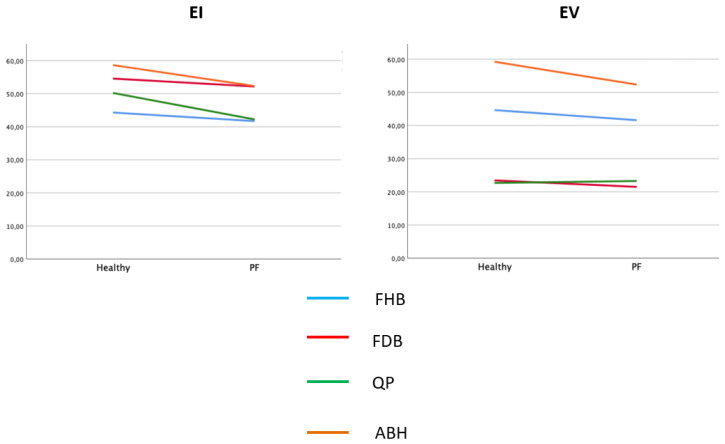
Mean EI and EV values for FHB, FDB, QP and AHB muscles.

**Table 1 healthcare-10-00526-t001:** Sociodemographic data of the sample.

Data	Plantar Fasciitis (*n* = 32)	Controls (*n* = 32)	*p*-Value Cases vs. Controls
Age, y	43.00 ± 11.00	31.00 ± 9.50	0.001
Weight, kg	76.00 ± 27.00	73.00 ± 25.50	0.028
Height, m	1.71 ± 6.3	1.70 ± 9.68	0.465
BMI, kg/m^2^	28.02 ± 5.56	24.69 ± 5.45	0.031

Abbreviations: BMI, body mass index.

**Table 2 healthcare-10-00526-t002:** Ultrasound imaging measurements of the intrinsic muscle EI and EV.

Measurement	Plantar Fasciitis (*n* = 32) Mean ± SD (95% CI)	Controls (*n* = 32) Mean ± SD (95% CI)	*p*-Value
FHB EI	41.30 ± 13.35 (36.4–46.1)	44.37 ± 9.77 (40.8–47.9)	0.297
FHB EV	41.19 ± 13.02 (36.4–45.8)	44.77 ± 10.14 (41.11–48.4)	0.225
FDB EI	52.02 ± 20.41 (44.6–59.3)	54.71 ± 12.26 (50.2–59.1)	0.527
FDB EV	21.35 ± 8.67 (18.2–24.4)	23.63 ± 6.33 (21.3–25.9)	0.236
QP EI	43.14 ± 19.37 (36.1–50.1)	50.94 ± 18.45 (44.2–57.5)	0.105
QP EV	22.94 ± 9.98 (19.3–26.5)	22.48 ± 6.66 (20.0–24.8)	0.829
AHB EI	52.06 ± 18.50 (45.3–58.7)	58.50 ± 13.07 (53.7–63.2)	0.113
AHB EV	52.12 ± 18.47 (45.4–58.7)	59.00 ± 13.37 (54.1–63.8)	0.093

Abbreviations: AHB, abductor hallucis brevis; CSA, cross-sectional area; FDB, flexor digitorum brevis; FHB, flexor hallucis brevis; QP, quadratus plantae.

## Data Availability

Data available under formal request.

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
