# Peer review of "Muscle Quality Assessment by Ultrasound Imaging of the Intrinsic Foot Muscles in Individuals with and without Plantar Fasciitis: A Case–Control Study"

_healthcare, 2022, doi:10.3390/healthcare10030526_

Round 1

Reviewer 1 Report

Excellent article very practical, which is an excellent addition to the literature on muscle function and activation within the foot. 

Author Response

  • In response to “Excellent article very practical, which is an excellent addition to the literature on muscle function and activation within the foot.”

[Response #1] Thank you very much for your comment. We are very excited that you like the manuscript.

Reviewer 2 Report

My opinion and suggestions are added as a attached file.

Regards

Author Response

  • In response to “The topic discussed in the manuscript seems to be very interesting in a clinical perspective.

[Response #1] Thank you very much for the comment. We are very happy that you like the manuscript and it´s clinical perspective.

  • In response to “In the introduction, the authors in details present the issues related to the main topic of the manuscript. The role of specific muscles involved in development of Plantar Fasciitis is quite well described – to get the full picture, it would also be worthy to mention the hypotheses related to the role of fascial disorders”.

[Response #2] Thank you very much for the comment. According to the reviewer idea, we add the following sentence “We hypothesized that patients diagnosed with PF might compensate the plantar fascia deficit with an overcompensation of the IFM with fascial and muscle architecture modifications due to a load adaptation process.”

  • In response to “In the methods section, the discrepancy between groups in terms of age, body weight and BMI is very striking. In my opinion that is significant methodological limitation. What is more, I could not find the data of the sampling frequency in the USG examination.”

[Response #3] Thank you very much for the suggestion. The linear transducer employed was 6 to 14 MHz linear transducer (L14-6NE). We employed the frequency of 10MHz.

  • In response to “A good solution is to introduce the Methodological Consideration section, where the manuscript limitations appears, but in my opinion it does not solve the problem of inappropriate selection of participants (as I have mentioned above). To sum up, the methodological limitations I have indicated, including significant differences between the groups, are impossible to correct at the present stage of the study. In my opinion, they seem to be a serious problem, despite the fact that the authors did not show any differences in the examined parameters.

[Response #4] Thank you very much for the suggestion and the comments in order to improve the quality of the manuscript. This limitation has been indicated into the methodological considerations section. In addition, in order to reduce bias for grouping-selection a lineal regression model was run with independent variable each sociodemographic variables and dependent each ultrasonography variables. The linear regression results showed non-significant differences for any of dependent variables checked (p = 0.102 to p = 0.789). Then, there is no bias of sociodemographic variables in neither group. In addition, the explanation has been added into the methods and results section.

  • In response to “The strong side of the manuscript is its bibliography, which is suited very well to the discussed issues. Moreover, it is based on current publications.”

[Response #5] Thank you very much for your time and your support.

Thanks for the Editor-in-Chief and Reviewers' comments to improve the quality of the manuscript.

Reviewer 3 Report

There are some minor issues that need to be addressed. Firstly, please indicate where the study was conducted and who were the volunteers participating in the study. Also, provide details regarding the number and date of getting the Ethics Committee approval. Also, I would suggest replacing the primary and secondary aim of the study as the reliability of the examination should be determined prior to its validity in a certain group of patients. Please provide legend to the figure to explain abbreviations.

Author Response

  • In response to “There are some minor issues that need to be addressed. Firstly, please indicate where the study was conducted and who were the volunteers participating in the study. Also, provide details regarding the number and date of getting the Ethics Committee approval. Also, I would suggest replacing the primary and secondary aim of the study as the reliability of the examination should be determined prior to its validity in a certain group of patients. Please provide legend to the figure to explain abbreviations.”

[Response #1] Thank you very much for the help and the support. The date and number of the Ethics Committee approval was included in the methods section. According to the reviewer, we switch the primary and secondary objectives with the following sentence “Therefore, the primary aim of the present study was to assess the intra-rater reliability of the echotexture variables. The secondary objective was to compare the EI and EV of the IFM between individuals with and without PF, using ultrasound imaging.”. Thank you very much, coinciding with the reviewer we think that the introduction section was improved. At last, legends was placed under the figures.

Thanks for the Editor-in-Chief and Reviewers' comments to improve the quality of the manuscript.

Round 2

Reviewer 2 Report

I still dissatisfied, that the control group, and main group are so different (weight and age), but I know that this is not so important. I accepted your corrections in the paper. I think that aim of your study is very interesting, and I hope that your conclusions may by helpful for practitioners. 

Regards